# A Unified View of Delta Parameter Editing in Post-Trained Large-Scale Models

## Abstract

Post-training has emerged as a crucial paradigm for adapting large-scale pre-trained models to various tasks, whose effects are fully reflected by delta parameters (i.e., the disparity between post-trained and pre-trained parameters). While numerous studies have explored delta parameter properties via operations like pruning, quantization, low-rank approximation, and extrapolation, a unified framework for systematically examining these characteristics has been lacking. In this paper, we propose a novel perspective based on Riemann sum approximation of the loss function to elucidate delta parameter editing operations. Our analysis categorizes existing methods into three classes based on their post-editing performance: competitive, decreased, and improved, explaining how they are expressed by the Riemann sum approximation term and how they alter the model performance. Extensive experiments on both visual and language models, including ViT, LLaMA 3, and Mistral, corroborate our theoretical findings. Furthermore, we introduce extensions to existing techniques like DARE and BitDelta, highlighting their limitations in leveraging the properties of delta parameters and reorganizing them into general expressions to enhance the applicability and effectiveness of delta parameter editing in post-trained models.

## 1 Introduction

With the remarkable success of large-scale pre-trained models, post-training has emerged as the de facto standard paradigm for effective adaptations to various tasks (Han et al., 2024; Xin et al., 2024; Dodge et al., 2020; Zhao et al., 2023). Conceptually, post-training optimizes the parameters of pre-trained backbone on task-specific data, endowing models with diverse abilities like visual recognition (Chen et al., 2022; Sandler et al., 2022), instruction following (Rafailov et al., 2023; Ethayarajh et al., 2024), and mathematical reasoning (Luo et al., 2023; Tong et al., 2024). It has been noted that the impact of post-training is fully manifested in the *delta parameters*, which are defined as the difference between parameters of pre-trained and post-trained models (Ilharco et al., 2023; Yu et al., 2024).

Due to the inherent correlations between delta parameters and post-training, significant efforts have been made to investigate the properties of delta parameters through various editing operations in recent years. For instance, studies like DARE (Yu et al., 2024) and DELLA-Merging (Deep et al., 2024) showed that models can achieve comparable performance with only a small fraction of delta parameters, highlighting their extreme redundancy. BitDelta (Liu et al., 2024) demonstrated that delta parameters could be quantized to 1 bit with modest performance compromise. Twin-Merging (Lu et al., 2024) and TIES-Merging (Yadav et al., 2023) discovered that most of the benefits of post-training can be retained after executing singular value decomposition and magnitude-based pruning on delta parameters. EXPO (Zheng et al., 2024) observed that cheaply extrapolating delta parameters with a suitable scaling factor can even enhance the performance. However, a comprehensive framework for systematically discussing delta parameter characteristics and theoretically explaining how different operations impact model performance remains lacking.

In this work, we make a pioneering effort to provide a unified view of delta parameter editing in post-trained large-scale models. We formulate the editing operations of delta parameters based on Riemann sum approximation of the loss of the edited model. By mathematically representing existing editing operations with the approximation term, we elucidate why certain operations result in

competitive, decreased, or improved performance. Specifically, we verify that: 1) methods such as DARE and DELLA-Merging can well keep the approximation term to zero through the random drop and rescale processes, ensuring equal loss between the edited and post-trained models and achieving competitive performance. 2) techniques including BitDelta, Twin-Merging, and TIES-Merging often result in decreased performance due to a positive approximation term introduced by quantization, low-rank approximation, and magnitude-based pruning; 3) EXPO-like methods can restrict the loss of the edited model to be less than that of the post-trained model by yielding a negative approximation term. To validate our theoretical analysis, extensive experiments are conducted on large-scale visual models (ViT (Radford et al., 2021)) and language models (LLaMA 3 (Dubey et al., 2024), and Mistral (Jiang et al., 2023)), and the results strongly support our analysis.

Besides understanding existing delta parameter editing techniques in the proposed view, we further present several extensions to provide more general formats. Firstly, we introduce a factor to handle the dropped parameters in DARE, effectively expanding methods like DARE. Secondly, we extend the scope of quantification-based methods like BitDelta, identifying a broader area for effective quantification beyond solely utilizing the average sum of delta parameters. Finally, we identify that extrapolation is not the key to the success of EXPO-like methods. Instead, we should determine whether to use extrapolation or interpolation based on the direction of the approximation term. Experimental results also demonstrate the effectiveness of the proposed extensions.

## 2  RELATED WORK

### 2.1  POST-TRAINING OF LARGE-SCALE MODELS

In recent years, with the rapid development of large-scale models, post-training has become an essential process for adapting the pre-trained backbone to a variety of tasks (Xin et al., 2024; Dodge et al., 2020; Zhao et al., 2023). Post-training realizes the adaptation via adjusting the pre-trained backbone's parameters through full fine-tuning (Dosovitskiy et al., 2021; Liu et al., 2021; Devlin et al., 2019; Radford et al., 2018) or parameter-efficient fine-tuning (He et al., 2023; Houlsby et al., 2019; Li & Liang, 2021; Hu et al., 2022; Han et al., 2024) algorithms. It is straightforward to conclude that the effectiveness of post-training can be perfectly denoted by the delta parameters, which represent the difference between post-trained and pre-trained parameters (Ilharco et al., 2023; Yu et al., 2024). Given the close correlations between delta parameters and the post-training process, investigating the properties of delta parameters becomes particularly important. In this paper, we present a novel perspective to illustrate delta parameter characteristics of post-trained models.

### 2.2  DELTA PARAMETER EDITING FOR POST-TRAINED MODELS

Existing delta parameter editing techniques can be generally categorized as three aspects according to their post-editing performance, including competitive, decreased, and improved performance.

**Delta Parameter Editing with Competitive Performance**. DARE (Yu et al., 2024) is a widely used approach to edit delta parameters without compromising the model performance. Technically, DARE can eliminate most (90% or even 99%) of the delta parameters with the random drop and rescale operations. Inspired by DARE, DELLA-Merging (Deep et al., 2024) presented a magnitude-aware drop to replace the random drop for achieving better performance, which ranks delta parameters by their magnitude and assigns higher dropout probabilities to those with lower ranks (i.e., corresponding to lower magnitudes). Yu et al. (2024) and Deep et al. (2024) explained that DARE and DELLA-Merging can work because they are able to approximate the original embeddings based on only a small fraction of delta parameters, thus maintaining the model performance.

**Delta Parameter Editing with Decreased Performance**. BitDelta (Liu et al., 2024) quantized delta parameters to only 1 bit according to the average magnitude scalar and sign bits. Twin-Merging (Lu et al., 2024) applied singular value decomposition (Klema & Laub, 1980) on delta parameters to extract exclusive knowledge for each specific task. TIES-Merging (Yadav et al., 2023) retained delta parameters with the largest magnitudes for reducing redundancy. All the above methods yield slightly worse results after executing the corresponding quantization, low-rank approximation, or pruning operations.

**Delta Parameter Editing with Improved Performance**. EXPO (Zheng et al., 2024) extrapolated delta parameters calculated by two relatively weaker models with an appropriate scaling factor to construct a stronger model, which can enhance the model performance.

It can be concluded that current approaches utilizes distinct operations for editing delta parameter, lacking a comprehensive analysis of whether these editing operations are suitable and why different operations cause various influence on the model performance. In this work, we make the first attempt to introduce a unified view of delta parameter editing in post-training, which is supported both theoretically and empirically.

## 3 PRELIMINARIES

### 3.1 NOTATIONS

**Delta Parameters During Post-Training**. Let $\boldsymbol{W}_{\text{PRE}} \in \mathbb{R}^{d \times k}$ denote the parameters of a pre-trained model, where $d$ and $k$ represent the output and input dimensions. A post-trained model with parameters $\boldsymbol{W}_{\text{POST}} \in \mathbb{R}^{d \times k}$ can be derived from the pre-trained backbone, yielding delta parameters $\Delta \boldsymbol{W} = \boldsymbol{W}_{\text{POST}} - \boldsymbol{W}_{\text{PRE}} \in \mathbb{R}^{d \times k}$. As delta parameters denote the alterations of parameters during the post-training process, investigating the characteristics of delta parameters can provide a deeper understanding of post-training.

**Delta Parameter Editing**. Let $f(\Delta \mathbf{W})$ represent the delta parameter editing function. The edited parameters $\Delta \widetilde{\boldsymbol{W}}_{\text{Edit}} = f(\Delta \mathbf{W})$ is then combined with $\mathbf{W}_{\text{PRE}}$ to obtain the final edited parameter $\mathbf{W}_{\text{Edit}} = \mathbf{W}_{\text{PRE}} + \Delta \widetilde{\boldsymbol{W}}_{\text{Edit}}$. Existing delta parameter editing methods can be categorized into three types based on their effects on model performance, i.e., competitive, decreased, and improved performance. These methods employ various techniques including pruning, quantization, low-rank approximation, and extrapolation. Notable works in this field include DARE, BitDelta, Twin-MERGING, TIES-Merging, and EXPO, which are investigated in this paper.

### 3.2 A UNIFIED VIEW OF DELTA PARAMETER EDITING

In this work, we introduce a unified view of delta parameter editing during the post-training process based on Riemann sum approximation. Specifically, we represent the changes caused by existing editing methods by $\Delta \widetilde{\boldsymbol{W}}$ and aim to investigate their effects on performance via analyzing the Riemann sum approximation term, which corresponds to the difference in loss made by the editing operation as follows,

$$
\Delta \mathcal{L} = \mathcal{L}(\boldsymbol{W}_{\text{POST}} + \Delta \widetilde{\boldsymbol{W}}) - \mathcal{L}(\boldsymbol{W}_{\text{POST}}) = \int_0^1 \nabla \mathcal{L}(\boldsymbol{W}_{\text{POST}} + t \Delta \widetilde{\boldsymbol{W}}) \cdot \Delta \widetilde{\boldsymbol{W}} \, dt
$$

$$
\approx \frac{1}{C} \sum_{c=0}^{C-1} \langle \nabla \mathcal{L}(\boldsymbol{W}_{\text{POST}} + \frac{c}{C} \Delta \widetilde{\boldsymbol{W}}), \Delta \widetilde{\boldsymbol{W}} \rangle = \frac{1}{C} \sum_{c=0}^{C-1} \langle \nabla \mathcal{L}^c, \Delta \widetilde{\boldsymbol{W}} \rangle,
\tag{1}
$$

where $\mathcal{L}(\boldsymbol{W}) : \mathbb{R}^{d \times k} \rightarrow \mathbb{R}$ denotes the loss function of a model with parameters $\boldsymbol{W} \in \mathbb{R}^{d \times k}$, $\nabla \mathcal{L}(\boldsymbol{W})$ is the gradient of the loss function at $\boldsymbol{W}$, and $\langle \cdot, \cdot \rangle$ denotes the Frobenius inner product. $C$ denotes the number of subdivisions of the interval $[0, 1]$. This expansion provides a linear approximation of the loss function in the neighborhood of $\boldsymbol{W}_{\text{POST}}$, allowing the analysis of the impact of parameter changes on the model performance. In most cases, the loss difference can reflect the influence on performance, with a positive value indicating deterioration, zero indicating stability, and a negative value indicating improvement. In section 4, section 5, and section 6, we respectively discuss editing operations that cause competitive, decreased, and improved performance, and derive the format of these operations when organizing them into the proposed unified paradigm.

To validate our theoretical analysis and the proposed extensions, we conducted experiments on LLaMA-3-8B-Instruct (Dubey et al., 2024), Mistral-7B-Instruct-v0.3 (Jiang et al., 2023), and ViT-B-32 (Radford et al., 2021). We evaluate text models on 8 tasks: 25-shot ARC Challenge (Clark et al., 2018), 5-shot GSM8K (Cobbe et al., 2021), 10-shot HellaSwag (Zellers et al., 2019), zero-shot HumanEval (Chen et al., 2021), zero-shot IFEval (Zhou et al., 2023), 5-shot MMLU (Hendrycks et al., 2020), zero-shot TruthfulQA (Lin et al., 2021), and zero-shot Winogrande (Sakaguchi et al., 2021),

and evaluate vision models on 3 tasks: DTD (Cimpoi et al., 2014), EuroSAT (Helber et al., 2019), and GTSRB (Stallkamp et al., 2011).

# 4 UNIFYING EDITING OPERATIONS WITH COMPETITIVE PERFORMANCE

As a widely-used approach for delta parameter editing, DARE (Yu et al., 2024) presents the random drop and rescale process to remove 90% or even 99% delta parameters without compromising the model performance. Following this line, many follow-up works have been proposed. For example, DELLA-Merging (Deep et al., 2024) modifies the drop operation in DARE from random to magnitude-aware. In this section, we select DARE for analysis because it is the most representative method among those that can retain the original model performance after editing delta parameters.

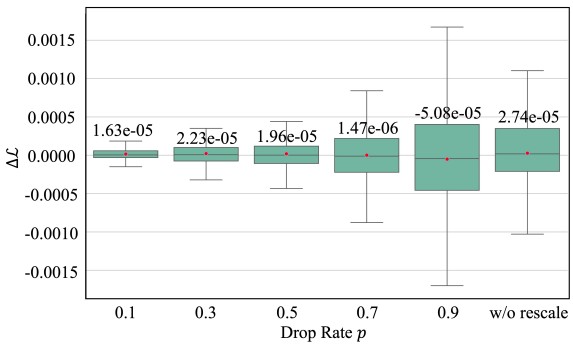

Figure 1: Validation of our theoretical derivation of DARE. The rightmost part labeled "w/o rescaling" represents the baseline.

## 4.1 EXPRESS DARE WITH APPROXIMATION TERM

Mathematically, the editing process of delta parameters in DARE is denoted by

$$W_{\text{DARE}} = W_{\text{POST}} + \Delta \widetilde{W}_{\text{DARE}} = W_{\text{PRE}} + \Delta W + \Delta \widetilde{W}_{\text{DARE}}$$

$$= W_{\text{PRE}} + 0 \cdot M \odot \Delta W + \frac{1}{1-p} \cdot (1 - M) \odot \Delta W = W_{\text{PRE}} + \frac{1}{1-p} \cdot (1 - M) \odot \Delta W, \quad (2)$$

where $p \in \mathbb{R}$ represents the drop rate and $\odot$ denotes the element-wise Hadamard product. $M \sim$ Bernoulli$(p, \Delta W) \in \mathbb{R}^{d \times k}$ is a mask matrix sampled from Bernoulli distribution according to $p$, whose shape is identical to that of $\Delta W$. From Equation (2), we can derive that

$$\Delta \widetilde{W}_{\text{DARE}} = \frac{p - M}{1-p} \odot \Delta W. \quad (3)$$

Referring to Equation (1), we obtain

$$\Delta \mathcal{L} \approx \frac{1}{C} \sum_{c=0}^{C-1} \sum_{i=1}^{d} \sum_{j=1}^{k} \frac{p - M_{ij}}{1-p} \cdot \Delta W_{ij} \cdot \nabla \mathcal{L}_{ij}^c$$

$$= \frac{1}{C} \sum_{c=0}^{C-1} \left( \frac{p}{1-p} \cdot \sum_{M_{ij}=0} \Delta W_{ij} \cdot \nabla \mathcal{L}_{ij}^c - \sum_{M_{ij}=1} \Delta W_{ij} \cdot \nabla \mathcal{L}_{ij}^c \right). \quad (4)$$

Due to the randomness of the drop operation in DARE, it is straightforward to deduce that

$$\sum_{M_{ij}=0} \Delta W_{ij} \cdot \nabla \mathcal{L}_{ij}^c = (1 - p) \cdot \sum_{i=1}^{d} \sum_{j=1}^{k} \Delta W_{ij} \cdot \nabla \mathcal{L}_{ij}^c,$$

$$\sum_{M_{ij}=1} \Delta W_{ij} \cdot \nabla \mathcal{L}_{ij}^c = p \cdot \sum_{i=1}^{d} \sum_{j=1}^{k} \Delta W_{ij} \cdot \nabla \mathcal{L}_{ij}^c. \quad (5)$$

Substituting Equation (5) into Equation (4), we derive

$$\Delta \mathcal{L} \approx (\frac{p}{1-p} \cdot (1 - p) - p) \cdot \frac{1}{C} \sum_{c=0}^{C-1} \sum_{i=1}^{d} \sum_{j=1}^{k} \Delta W_{ij} \cdot \nabla \mathcal{L}_{ij}^c = 0. \quad (6)$$

To this end, we can conclude that after editing delta parameters with DARE, the loss $\mathcal{L}(\boldsymbol{W}_{\text{DARE}})$ remains identical to $\mathcal{L}(\boldsymbol{W}_{\text{POST}})$, explaining why DARE can achieve competitive performance even most delta parameters are eliminated.

To verify the above analysis, we used the DARE method to construct models on LLaMA3-8B-Instruct and computed the approximation term on the GSM8K dataset. The results are shown in Figure 1. We used the scenario where 50% of the delta parameters were masked without rescaling as a reference (the rightmost part of the figure). As can be seen, models with DARE constructed consistently achieved lower average loss, and with a smaller drop rate, the approximation term calculated across different parts of the model remained relatively small. This validates our theoretical derivation above.

## 4.2 EXTENSION OF DARE

We further present a more general format of delta parameter editing operations that can achieve competitive performance. In particular, instead of dropping delta parameters, we introduce a term $k$ to adjust them and rescale the remaining ones with $(1 - k \cdot p)/(1 - p)$. Similar to the deduction in Equation (2) to Equation (6), we obtain

$$
\boldsymbol{W}_{\text{COMP}} = \boldsymbol{W}_{\text{PRE}} + \Delta \boldsymbol{W} + \Delta \widetilde{\boldsymbol{W}}_{\text{COMP}} = \boldsymbol{W}_{\text{PRE}} + k \cdot \boldsymbol{M} \odot \Delta \boldsymbol{W} + \frac{1 - k \cdot p}{1 - p} \cdot (1 - \boldsymbol{M}) \odot \Delta \boldsymbol{W},
$$

$$
\Delta \widetilde{\boldsymbol{W}}_{\text{COMP}} = \frac{(k - 1)(\boldsymbol{M} - p)}{1 - p} \odot \Delta \boldsymbol{W},
$$

$$
\Delta \mathcal{L} \approx \frac{1}{C} \sum_{c=0}^{C-1} \left( \frac{p \cdot (1 - k)}{1 - p} \cdot \sum_{M_{ij}=0} \Delta W_{ij} \cdot \nabla \mathcal{L}_{ij}^{c} + (k - 1) \cdot \sum_{M_{ij}=1} \Delta W_{ij} \cdot \nabla \mathcal{L}_{ij}^{c} \right)
$$

$$
= \left( \frac{p \cdot (1 - k)}{1 - p} \cdot (1 - p) + (k - 1) \cdot p \right) \cdot \frac{1}{C} \sum_{c=0}^{C-1} \sum_{i=1}^{d} \sum_{j=1}^{k} \Delta W_{ij} \cdot \nabla \mathcal{L}_{ij}^{c} = 0.
$$

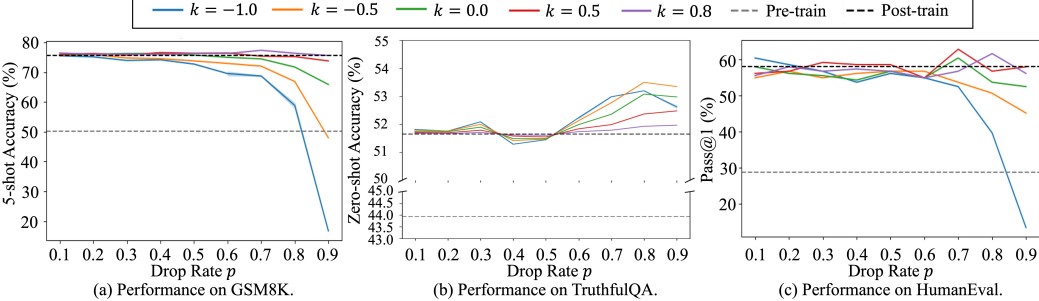

Figure 2: The performance of LLaMA3-8B-Instruct on the GSM8K, TruthfulQA, and HumanEval datasets under varying $p$ and $k$.

It has been verified that $\langle \nabla \mathcal{L}(\boldsymbol{W}_{\text{POST}}), \Delta \widetilde{\boldsymbol{W}}_{\text{COMP}} \rangle$ equals 0, which indicates the validity of the proposed format. Note that in DARE, the drop operation can be realized by setting $k$ to 0. Thus, our format is an extension of DARE with broader settings of $k$.

We conducted validation experiments for the extension of DARE on LLaMA3-8B-Instruct and ViT-B-32. The results are shown in Figure 2 and Figure 3. Specifically, on four representative text datasets—GSM8K, TruthfulQA, and HumanEval, when both the rescale rate $k$ and sign change rate $kp$ are small (e.g., less than 0.5), the performance of our adjusted model is very close to that of the original post-trained model and significantly outperforms the pre-trained model. Regarding the weight scalar $k$ introduced in our extension, we observed that, compared to the setting where $k = 0$ (which reverts to the original DARE configuration), using $k \neq 0$ generally yields competitive performance across different datasets. This demonstrates the effectiveness of our extension. For

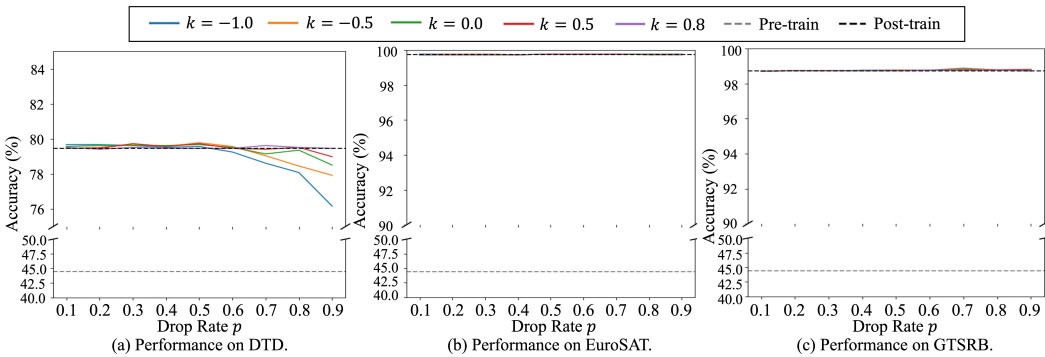

Figure 3: The performance of ViT-B-32 on the DTD, EuroSAT, and GTSRB datasets under varying $p$ and $k$.

the ViT model, the results on the DTD, EuroSAT, and GTSRB datasets are more consistent with our expectations. Regardless of the rescale and sign change rates, the performance of the adjusted model is almost identical to that of the original post-trained model.

### 4.3 Further Discussions on DARE

Yu et al. (2024) and Deep et al. (2024) claim that DARE and DELLA-Merging are effective because the random drop of delta parameters ensures an approximation of the original embeddings, thereby preserving model performance. However, according to our established view, we argue that random drop of delta parameters is a sufficient but not necessary condition for maintaining model performance. Furthermore, we contend that ensuring randomness in the element-wise product of delta parameters and approximation term is the necessary and sufficient condition.

To verify the above analysis, we conduct two experiments on GSM8K dataset. First, we disrupt the randomness of the delta parameter drop operation by multiplying all negative delta parameters by $k$ and all positive delta parameters by $(1 - k \cdot p)/(1 - p)$. The results are shown in the middle column of Table 1, illustrating that the model performance remains intact. This validates that the randomness of the delta parameter dropout operation is a sufficient but not necessary condition for maintaining model performance. Furthermore, we disrupt the randomness of the dropout operation on the approximation term by multiplying all negative products by $k$ and all positive products by $(1 -$

| k | Random | Biased $\Delta W$ | Biased $\Delta W \cdot \nabla L$ |
|---|--------|-------------------|-----------------------------------|
| 0.5 | 76.35 | 74.15 | 0.0 |
| 0.7 | 75.89 | 75.36 | 0.0 |
| 0.9 | 76.19 | 76.04 | 26.76 |
| 1.1 | 75.89 | 75.59 | 0.15 |
| 1.3 | 75.36 | 74.91 | 0.0 |
| 1.5 | 75.59 | 74.83 | 0.0 |

Table 1: Validation of the discussion on DARE. The leftmost column shows the random drop in DARE. The middle column illustrates the approach of multiplying all negative delta parameters by $k$ and all positive delta parameters by $\frac{1-k \cdot p}{1-p}$. The rightmost column demonstrates the method of first calculating the product of delta parameters and gradients, and then multiplying all negative products by $k$ and all positive products by $\frac{1-k \cdot p}{1-p}$.

$k \cdot p)/(1 - p)$. The results, as depicted in the rightmost of Table 1, show a significant decline in model performance. This validates that the randomness of the dropout operation on the product of delta parameters and approximation term is a necessary and sufficient condition for maintaining model performance.

## 5 Unifying Editing Operations with Decreased Performance

This section discusses three delta parameter editing operations that incur reduced results, including quantization, low-rank approximation, and pruning. We respectively choose BitDelta (Liu et al., 2024), Twin-Merging (Lu et al., 2024), and TIES-Merging (Yadav et al., 2023) as typical works.

## 5.1 EXPRESS BITDELTA WITH APPROXIMATION TERM

BitDelta quantizes delta parameters down to 1 bit, utilizing the sign bit matrix and a high-precision scalar, where the latter is computed by the average magnitude of delta parameters. Specifically, BitDelta can be represented by

$$\boldsymbol{W}_{\text{BitDelta}} = \boldsymbol{W}_{\text{POST}} + \Delta\widetilde{\boldsymbol{W}}_{\text{BitDelta}} = \boldsymbol{W}_{\text{PRE}} + \Delta\boldsymbol{W} + \Delta\widetilde{\boldsymbol{W}}_{\text{BitDelta}}$$

$$= \boldsymbol{W}_{\text{PRE}} + \frac{1}{d \cdot k}\sum_{i=1}^{d}\sum_{j=1}^{k}|\Delta W_{ij}| \cdot \text{Sign}(\Delta\boldsymbol{W}) = \boldsymbol{W}_{\text{PRE}} + \text{AVG}(|\Delta\boldsymbol{W}|) \cdot \text{Sign}(\Delta\boldsymbol{W}), \quad (7)$$

where $|\cdot|$ denotes the operation of taking magnitudes. $\text{AVG}(|\Delta\boldsymbol{W}|)$ represents the average magnitude of $\Delta\boldsymbol{W}$. Since $\Delta\boldsymbol{W} = |\Delta\boldsymbol{W}| \odot \text{Sign}(\Delta\boldsymbol{W})$, based on Equation (7), we can further obtain

$$\Delta\widetilde{\boldsymbol{W}}_{\text{BitDelta}} = (\text{AVG}(|\Delta\boldsymbol{W}|) - |\Delta\boldsymbol{W}|) \odot \text{Sign}(\Delta\boldsymbol{W}). \quad (8)$$

Based on Equation (1), we get

$$\Delta\mathcal{L} \approx \frac{1}{C}\sum_{c=0}^{C-1}\sum_{i=1}^{d}\sum_{j=1}^{k}(\text{AVG}(|\Delta\boldsymbol{W}|) - |\Delta W_{ij}|) \cdot \text{Sign}(\Delta W_{ij}) \cdot \nabla\mathcal{L}_{ij}^{c}. \quad (9)$$

Though $\sum_{i=1}^{d}\sum_{j=1}^{k}((\text{AVG}(|\Delta\boldsymbol{W}|) - |\Delta W_{ij}|) = d \cdot k \cdot \text{AVG}(|\Delta\boldsymbol{W}|) - \sum_{i=1}^{d}\sum_{j=1}^{k}|\Delta W_{ij}| = 0$, it is hard to conclude that Equation (9) equals 0 due to the multiplication of $\text{Sign}(\Delta W_{ij}) \cdot \nabla\mathcal{L}_{ij}^{c}$.

## 5.2 EXPRESS TWIN-MERGING AND TIES-MERGING WITH APPROXIMATION TERM

Twin-Merging employs singular value decomposition on delta parameters to derive task-specific knowledge. TIES-Merging preserves delta parameters with the highest magnitudes to minimize redundancy. Their computation processes are

$$\boldsymbol{W}_{\text{Twin}} = \boldsymbol{W}_{\text{POST}} + \Delta\widetilde{\boldsymbol{W}}_{\text{Twin}} = \boldsymbol{W}_{\text{PRE}} + \Delta\boldsymbol{W} + \Delta\widetilde{\boldsymbol{W}}_{\text{Twin}} = \boldsymbol{W}_{\text{PRE}} + \boldsymbol{U}_r\boldsymbol{\Sigma}_r\boldsymbol{V}_r^{T},$$

$$\boldsymbol{W}_{\text{TIES}} = \boldsymbol{W}_{\text{POST}} + \Delta\widetilde{\boldsymbol{W}}_{\text{TIES}} = \boldsymbol{W}_{\text{PRE}} + \Delta\boldsymbol{W} + \Delta\widetilde{\boldsymbol{W}}_{\text{TIES}} = \boldsymbol{W}_{\text{PRE}} + \boldsymbol{M} \odot \Delta\boldsymbol{W}, \quad (10)$$

where rank $r \leq \min(d, k)$ denotes the number of linearly independent columns (or rows) in $\Delta\boldsymbol{W} = \boldsymbol{U}\boldsymbol{\Sigma}\boldsymbol{V}^{T}$. $\boldsymbol{U}_r \in \mathbb{R}^{d\times r}$ consists of the first $r$ columns of $\boldsymbol{U}$ (whose columns are the left singular vectors of $\Delta\boldsymbol{W}$). $\boldsymbol{\Sigma}_r$ is the $r \times r$ diagonal matrix containing the top $r$ singular values. $\boldsymbol{V}_r \in \mathbb{R}^{k\times r}$ includes the first $r$ columns of $\boldsymbol{V}$ (whose columns are the right singular vectors of $\Delta\boldsymbol{W}$). $\boldsymbol{M} \in \mathbb{R}^{d\times k}$ is a binary mask matrix where an entry of 1 indicates that the corresponding delta parameter is among the top-$n$ percent in magnitude. $n$ is the proportion of delta parameters to be retained. According to Equation (10), we derive

$$\Delta\widetilde{\boldsymbol{W}}_{\text{Twin}} = \boldsymbol{U}_r\boldsymbol{\Sigma}_r\boldsymbol{V}_r^{T} - \Delta\boldsymbol{W},$$

$$\Delta\widetilde{\boldsymbol{W}}_{\text{TIES}} = \boldsymbol{M} \odot \Delta\boldsymbol{W} - \Delta\boldsymbol{W} = -\neg\boldsymbol{M} \odot \Delta\boldsymbol{W}, \quad (11)$$

where $\neg\boldsymbol{M}$ is the element-wise NOT operation. Based on Equation (1), we get

$$\Delta\mathcal{L}_{\text{Twin}} \approx \frac{1}{C}\sum_{c=0}^{C-1}\sum_{i=1}^{d}\sum_{j=1}^{k}(\boldsymbol{U}_r\boldsymbol{\Sigma}_r\boldsymbol{V}_r^{T}{}_{ij} - \Delta W_{ij}) \cdot \nabla\mathcal{L}_{ij}^{c},$$

$$\Delta\mathcal{L}_{\text{TIES}} \approx -\frac{1}{C}\sum_{c=0}^{C-1}\sum_{i=1}^{d}\sum_{j=1}^{k}\neg M_{ij} \cdot \Delta W_{ij} \cdot \nabla\mathcal{L}_{ij}^{c}. \quad (12)$$

We exploit the value of the approximation term through experiments. Models were constructed using LLaMA3-8B-Instruct, and the approximation term was calculated on the GSM8K dataset. As shown in Figure 4, the approximation losses are consistently greater than zero, which aligns with the observed performance degradation on the GSM8K dataset.

## 5.3 EXTENSION OF BITDELTA

We also extend the applicability of Bit-Delta by offering a more general form. Firstly, in addition to selecting the signs of delta parameters, we hypothesize that the effectiveness of BitDelta may stem from its choice of a holistic statistic that reflects the properties of the delta parameters. Specifically, BitDelta utilizes the average magnitude of delta parameters to achieve the best approximation error in the $L_2$ norm. To validate this, we conduct an experiment where we alter the holistic statistic selected by BitDelta, introducing varying degrees of noise to the average value. As illustrated in the "Degenerate" line of Figure 6, using the true aver-

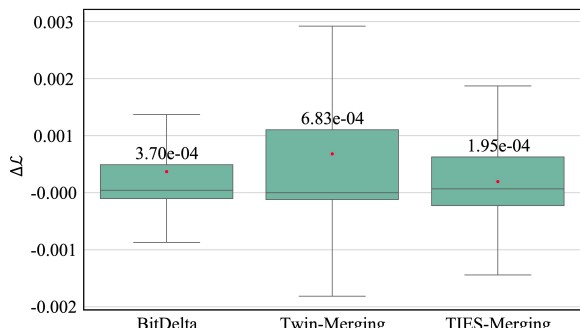

Figure 4: Validation of our theoretical analysis on operations with decreased performance.

age magnitude of the delta parameters (represented by the star marker in Figure 6, corresponding to BitDelta) yields nearly optimal performance on GSM8K. However, the performance on IFEval is somewhat anomalous, which may caused by the difficulty of instruction-following tasks and we will address this in future work. The performance changes along the degenerate line are quite steep, and slight modifications to this average value may result in a degradation of model performance.

Secondly, instead of using a single value, we sample delta parameter magnitude matrices from both standard normal and uniform distributions, with the average magnitude serving as the mean. The experimental results, as depicted in Figure 6, demonstrate that even when these parameters are randomly sampled from distributions, the model performance remains on par with a statistic value used in BitDelta. This further underscores the significance of selecting an appropriate holistic statistic for the delta parameters.

Finally, while preserving the relative magnitude relationships of delta parameters, we enhance the effectiveness of BitDelta by employing multiple bits. Specifically, we divide the delta parameters into $M$ blocks based on their magnitude, from smallest to largest. Each block is then represented by the average value of the delta

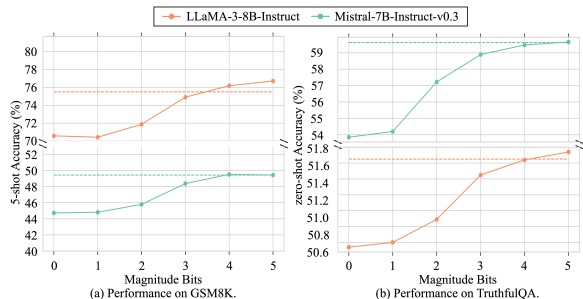

Figure 5: Effectiveness of increasing the number of bits in BitDelta. The left subplot shows the performance of LLaMA3-8B-Instruct and Mistral-7B-Instruct-v0.3 on the GSM8K dataset as the number of bits increases. The right subplot shows the performance on the TruthfulQA dataset. In each subplot, we use the dashed line to represent the performance of the original post-trained model.

parameters within that block. When $M = 1$, this approach corresponds to BitDelta, and when $M$ equals the total number of parameters in the model, it degenerates to the original post-trained model. The number of bits used is given by $\log_2 M$. As shown in Figure 5, increasing the number of bits significantly improves the model performance. When the number of bits is 4, the performance already surpasses that of the original post-trained model. This again highlights the redundancy in the delta parameters and demonstrates the potential for further advancements by expanding the bit representation in BitDelta.

## 6 Unifying Editing Operations with Improved Performance

EXPO (Zheng et al., 2024) is a recent method to extrapolate delta parameters, which can boost LLMs' alignment. This section chooses EXPO as the representative approach for illustration.

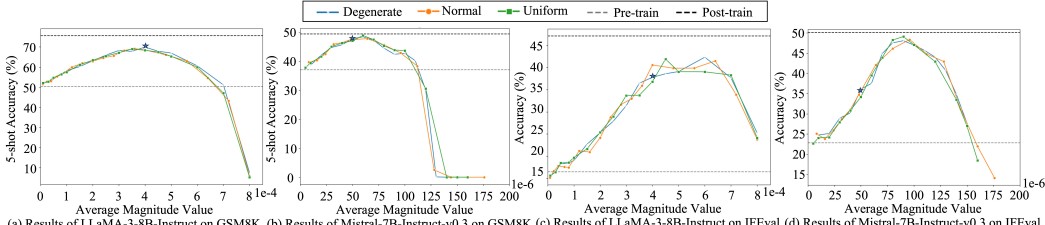

(a) Results of LLaMA-3-8B-Instruct on GSM8K. (b) Results of Mistral-7B-Instruct-v0.3 on GSM8K.(c) Results of LLaMA-3-8B-Instruct on IFEval.(d) Results of Mistral-7B-Instruct-v0.3 on IFEval.

Figure 6: Validation of the extension of BitDelta. The stars indicate the mean value of the delta parameters and the corresponding performance for the original BitDelta.

### 6.1 EXPRESS EXPO WITH APPROXIMATION TERM

Technically, EXPO first computes delta parameters between an aligned model and its initial fine-tuning checkpoints, and then extrapolates delta parameters with a suitable scaling factor for obtaining a better-aligned model. The calculation procedure is

$$\boldsymbol{W}_{\text{EXPO}} = \boldsymbol{W}_{\text{POST}} + \Delta\widetilde{\boldsymbol{W}}_{\text{EXPO}} = \boldsymbol{W}_{\text{PRE}} + \Delta\boldsymbol{W} + \Delta\widetilde{\boldsymbol{W}}_{\text{EXPO}} = \boldsymbol{W}_{\text{PRE}} + \Delta\boldsymbol{W} + \alpha\Delta\boldsymbol{W}, \quad (13)$$

where $\alpha$ controls the extrapolation length. Based on Equation (13), we derive

$$\Delta\widetilde{\boldsymbol{W}}_{\text{EXPO}} = \alpha\Delta\boldsymbol{W}. \quad (14)$$

Referring to Equation (1), we obtain

$$\Delta\mathcal{L}_{\text{EXPO}} \approx \frac{\alpha}{C} \cdot \sum_{c=0}^{C-1} \sum_{i=1}^{d} \sum_{j=1}^{k} \Delta W_{ij} \cdot \nabla\mathcal{L}_{ij}^{c}. \quad (15)$$

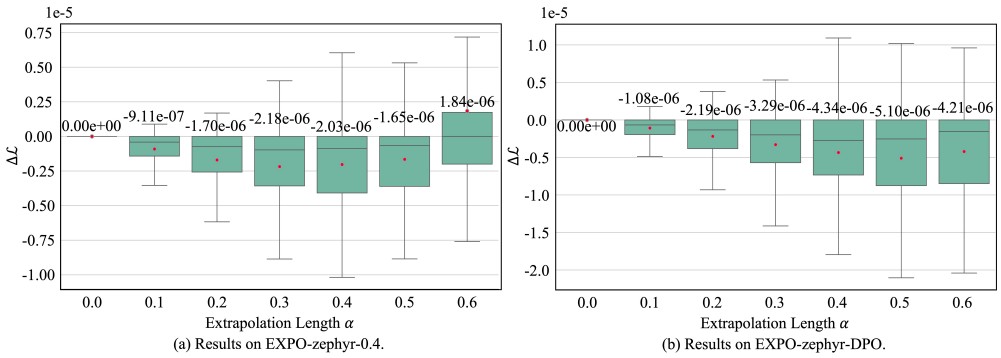

(a) Results on EXPO-zephyr-0.4.

(b) Results on EXPO-zephyr-DPO.

Figure 7: Validation of our theoretical analysis of EXPO. we can observe that the approximation term first decreases and then increases as alpha changes, indicating that optimal performance is achieved at the trough.

An intuitive explanation for the improvements that EXPO achieves is that the DPO/RLHF training process of these models is suboptimal, which leads to the direction of loss reduction (the negative gradient) still aligning with the direction of the delta parameters, causing Equation (15) to be negative. Consequently, the loss of the edited model is lower than that of the original post-training model, resulting in enhanced performance. We validated the aforementioned hypothesis on Zephyr-7B. Specifically, we conducted experiments using the EXPO-trained Zephyr-7B-DPO-Full and Zephyr-0.4 models. We calculated the gradient of the models using DPO loss on the evaluation set of UltraFeedback (Cui et al., 2024). As shown in Figure 7, when $\alpha$ is relatively small, the value of the loss approximation term gradually decreases, reflecting that the model is indeed suboptimal. Moving further in this direction decreases the loss and improves performance accordingly. However, as $\alpha$ increases, the loss term gradually increases until it exceeds zero, which is consistent with the observation in EXPO that there is an optimal value for $\alpha$.

### 6.2 FUTHER DISCUSSIONS ON EXPO

EXPO claims that extrapolating delta parameters leads to better models. However, based on the derivation in Equation (15), we believe that whether to use extrapolation or interpolation primarily depends on the direction of the approximation term (which is influenced by the specific data). Specifically, for LLaMA3-8B-Instruct, we uniformly selected $\alpha$ in the range of -1.0 to 1.0 at intervals of 0.1, performing both interpolation and extrapolation of the model's delta parameters. As show in Figure 8, on most datasets, interpolation outperformed extrapolation, except for the IFEval dataset, where extrapolation significantly improved performance. This confirms that whether to interpolate or extrapolate is not a fixed formula but depends on the specific data.

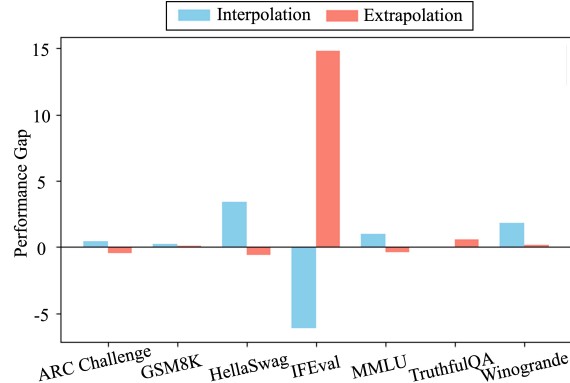

Figure 8: Comparison of Extrapolation and Interpolation Performance on LLaMA3-8B-Instruct. The performance gap represents the difference between the model's performance after extrapolation or interpolation and the original performance.

## 7 CONCLUSION AND DISCUSSIONS

Post-training is a core step in the training of large models. In recent years, significant efforts have been directed towards editing the delta parameters of post-training to achieve improvements in either performance or efficiency. However, while previous work has shown some effectiveness, the complexity of large model parameters has led to a fragmented understanding of delta parameter editing, with different studies focusing on different aspects of its effectiveness, lacking a unified perspective.

In this paper, we provide a unified perspective on the previous work related to post-training delta parameter editing using Riemann sum approximation. We find that the changes in model capability after altering the delta parameters essentially depend on the changes in the approximation term of Riemann sum approximation. Specifically, when the approximation term remains unchanged, the overall loss of the model remains stable, and thus the overall performance of the model also remains largely unchanged. When the approximation term decreases, the model's performance improves, and when the approximation term increases, the model's performance degrades.

Our work offers a concise, unified, and powerful explanation for almost all previous work in the field of post-training delta parameter editing. We validate our hypothesis through numerical experiments. From our conclusions, several potential applications emerge for future work in this direction: (1) Model Quantization: By finding an edit that sets the approximation term to zero while using lower precision, we can achieve nearly lossless compression of the model. (2) Model Enhancement: By directly controlling the approximation term, we can enhance the model's capabilities without additional training data. (3) Post-training Mechanism Analysis: Since the model's capability remains almost unchanged when the approximation term is zero, we can construct more concise post-training delta parameters. This simplifies the parameter changes during the post-training phase, enabling a more effective analysis of the parameter mechanisms in this stage.

Additionally, our work highlights a critical observation: the analysis of parameter changes during the post-training phase should not be limited to specific parameters, such as knowledge neurons, but should consider the overall distribution of parameters. This is because the key constraint of the approximation term being zero does not depend on the changes in a specific parameter during post-training but requires a comprehensive consideration of all parameter deltas. This suggests that trying to infer the impact on the global model parameters from changes in a single or a few local parameters is likely futile.

## REPRODUCIBILITY STATEMENT

We guarantee the reproducibility of our algorithm by providing the implementation code for download in the supplementary materials.

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
