# OpenReview forum: "A Unified View of Delta Parameter Editing in Post-Trained Large-Scale Models"
_ICLR.cc/2025/Conference — ICLR 2025 Conference Withdrawn Submission_

### Official Review · Reviewer_jdtS · 2024-10-28

**Soundness:** 2
**Presentation:** 1
**Contribution:** 1
**Rating:** 3
**Confidence:** 4

**Summary:**

The authors propose a framework using Riemann sum approximation to analyze delta editing (pruning, compression ..) methods based on their effects on model loss.

**Strengths:**

1. Delta parameter editing is an important topic in LLM efficiency.

**Weaknesses:**

The authors attempt to cover a broad scope in their analysis, but the exploration is often shallow, with certain aspects appearing incremental or inaccurate.
1. Assumptions weaken the generalizability and accuracy of their conclusions:

    a.   In Equation (4), the authors address the randomness in DARE [1] by asserting that “it is straightforward to deduce that,” leading to Equation (5). This is wrong as this need to rely on $\Delta W_{ij}$ and $\Delta \mathcal{L}_{ij}$ being uniformly distributed, which is generally not the case, making the equality invalid.

    b. In BitDelta [2] , the authors state that it is difficult to conclude that Equation (9) equals zero due to the interaction between $\text{sign}(\Delta W_{ij}) \Delta \mathcal{L_{ij}}$. This approach is inconsistent, as they assume uniformity of $\Delta W_{ij}$ and $\Delta \mathcal{L}_{ij}$  when proving Equation (5) for DARE, but not for BitDelta in Equation (9). If the uniformity also assumed for BitDelta, the $\mathcal{L}=0$ should hold. Consequently, the conclusion that BitDelta performs worse than DARE is questionable.

2. Question on EXPO[3]:

    a. Incremental Analysis and Use of EXPO Framework:  The paper uses the similar framework as EXPO, particularly referencing Equation (2) in EXPO, where EXPO use first-order Taylor Expansion with an alignment objective (which functions similarly to the loss used in the current paper). This similarity can be seen as an incremental extension rather than a substantial innovation.

    b. Claim on Gradient Correlation with Delta Parameter: In section 2.2 of EXPO, EXPO already established that the success of their approach depends on a positive correlation between the gradient and the delta parameter, highlighting a direct relationship with the approximation term. Given this established finding, it appears that the new paper is building on known results rather than offering a novel insight in this area.

   c. Claims on EXPO Limitations: According to EXPO, extrapolation can improve performance when a positive correlation between the gradient and the delta parameter, a point that the new paper seems to question. However, if EXPO already addressed this with clear justification, the claim in the current paper may not hold strong novelty or accuracy. If they misunderstand EXPO’s stance on extrapolation, it could weaken their argument about EXPO’s limitations.

3.  I feel the logic in DARE section is unclear. First, the extension of DARE lacks a clear connection to the theorem in Section 4.1, making the motivation for introducing k unclear. I though authors will give motivation in Section 4.3, but cannot found it. Also, in section 4.3, the authors claim that DARE overlooks delta loss. However, the original DARE analysis of random pruning considers both delta parameters and the input. Specifically, the authors’ analysis in Equation (4) focuses on delta parameters and delta loss, and due to the linear approximation, delta loss can be proportional to the input x. This resemblance to DARE makes it inappropriate to claim that DARE disregards delta loss (represented by x in DARE’s case).

4. I also find the logic in analyzing BitDelta [2] unclear. Similar to DARE, the motivation for introducing noise to mean magnitude lacks a clear connection to the theorem in Section 5.1. Additionally, the original BitDelta [2] already demonstrates that calibrating the scaling factors can improve performance as their contribution. This resemblance to BitDelta makes it inappropriate to claim that BitDelta overlooks this issue, which limits the novelty of the current approach.



[1]  Language Models are Super Mario: Absorbing Abilities from Homologous Models as a Free Lunch

[2]  BitDelta: Your Fine-Tune May Only Be Worth One Bit

[3] Weak-to-strong extrapolation expedites alignment.

**Questions:**

see weakness

---

### Official Review · Reviewer_Qn2S · 2024-11-03

**Soundness:** 1
**Presentation:** 2
**Contribution:** 2
**Rating:** 3
**Confidence:** 5

**Summary:**

This paper provides a unified view of weight-space editing methods (a.k.a. model merging) through the lens of approximated loss difference. The authors categorize existing methods into three classes -- maintained performance, increased performance, and decreased performance, and generalize the existing methods by analyzing the crucial factors in the proposed loss difference approximation framework.

**Strengths:**

- Present a novel unified view of existing model merging methods that is lacking so far.
- The proposed Riemann sum approximation of loss difference based analysis is interesting and gives insights for future work on theoretical understanding of model merging.

**Weaknesses:**

- **Eq. (1), the main proposed theoretical framework in this work, has a severe technical flaw**.
  - Specifically, the authors analyze $\Delta L = L(W_{POST} + \Delta \tilde{W})-L(W_{POST})$ to discuss the performance of existing model merging methods.
  - However, all the existing model merging methods mentioned in this work such as DARE [1], TIES-Merging [2], BitDelta [3], and so on, applied the edited delta parameter to the $W_{PRE}$ rather than $W_{POST}$.
  - Therefore, the desired analysis should be conducted on the loss term such as $L(W_{PRE}+\Delta \tilde{W})-L(W_{POST})$ or $L(W_{PRE}+\Delta \tilde{W})-L(W_{PRE})$ rather than the current form ($L(W_{POST}+\Delta \tilde{W})-L(W_{POST})$) to make any claims on the final downstream performance of existing merging methods.
- **Limited contributions**
  - Although the authors provide some generalization of existing methods, e.g., multiplies a magnitude hyperparameter $k$ in DARE framework, the novelty and innovativeness of these generalizations are too limited and the implications are also not surprising and uninformative. It seems like just reporting a result from engineering. Presenting more rigorous generalizations and providing profound implications from the proposed unified framework will improve the quality of this work significantly.
- **Unreasonable experiment setup**
  - Regarding the EXPO [4] method, the authors claim that the relative effectiveness of extrapolation and interpolation depends on the dataset, which shows the performances of interpolation and extrapolation over some NLP downstream tasks.
  - However, the motivation of EXPO is focused on the alignment for enhancing the instruction-following capability of large language models, and the authors should conduct the experiment about EXPO on that kind of benchmark such as AlpacaEval 2.0 adopted in the EXPO paper.
- **Bad presentation and validity of claims**
  - The quality of some presentations is not good enough for the purpose of publication. For example, see Figure 3. It is much better to omit the pre-train models' performance here to highlight the more important parts -- a comparison between varying $k$ values.
  - Moreover, although the authors make an argument based on some bar plots (Figure 1, Figure 4, Figure 7) they state the differences have some trend, and the absolute scale is too small among the comparison participants, which raises concerns about the statistical significance.



> Reference
1. Language Models are Super Mario: Absorbing Abilities from Homologous Models as a Free Lunch, Yu et al. 2024
2. TIES-Merging: Resolving Interference When Merging Models, Yadav et al. 2023
3. BitDelta: Your Fine-Tune May Only Be Worth One Bit, Liu et al. 2024
4. Weak-to-Strong Extrapolation Expedites Alignment, Zheng et al. 2024

**Questions:**

See the weaknesses section.
Please let the reviewer know if there is any misunderstanding about the paper.

---

### Official Review · Reviewer_mKaZ · 2024-11-05

**Soundness:** 1
**Presentation:** 3
**Contribution:** 1
**Rating:** 1
**Confidence:** 4

**Summary:**

Authors propose using an approximation term to evaluate various methods to compress the model. In particular, authors use Riemann sum to establish the connection between delta W and delta L. Authors discuss different cases where approximation term (delta L) is equal to, larger than, or smaller than 0.

**Strengths:**

1. The paper is clearly written.

**Weaknesses:**

1. The derivation in 4.1 is mathematically trivial due to this Riemann sum assumption. Either the locally constant assumption of Riemann sum is too strong or the expectation term derived in (5) is too strong. The math shows delta L is 0 regardless of p. If p=0.999, should the loss still be zero? In addition, I cannot see a connection between 4.1’s experiment and theory. The theory shows L is zero.
2. For the same reason, the math derivation in 4.2 is trivial. Adding a k does not have any effect on the proof.
3. Authors derived delta L in section 5 (larger than 0) and section 6 (smaller than 0), but there is no theoretical implication about why delta L in section 5 is positive and why delta L in section 6 is negative. The experiment results are empirical and whether it’s positive or negative has already been studied in BitDelta and EXPO.
4. A clear contradiction is: in section 4 when delta L is derived as zero, the experiment’s delta is 1e-5. While in section 6 when delta L is derived as non-zero, the absolute value of delta L is 1e-6, which is an order of magnitude smaller than the loss that is derived as zero. The experiment results also imply that the derivation in section 4 is false.
5. Overall I don’t see any value in the math derivation of this paper. The experiment part is also mostly expected after reading the paper that the respective section is referring to.

**Questions:**

see weaknesses

---

### Note · Authors · 2024-11-22

I have read and agree with the venue's withdrawal policy on behalf of myself and my co-authors.